# Effect of the Trunk and Upper Limb Passive Stabilization on Hand Movements and Grip Strength Following Various Types of Strokes—An Observational Cohort Study

**DOI:** 10.3390/brainsci12091234

**Published:** 2022-09-13

**Authors:** Anna Olczak, Aleksandra Truszczyńska-Baszak, Adam Stępień, Katarzyna Bryll

**Affiliations:** 1Rehabilitation Clinic, Military Institute of Medicine, 04-141 Warsaw, Poland; 2Faculty of Rehabilitation, Józef Piłsudski University of Physical Education, 00-968 Warsaw, Poland; 3Neurological Clinic, Military Institute of Medicine, 04-141 Warsaw, Poland; 4Outpatient Clinic, Military Institute of Medicine, 05-119 Legionowo, Poland

**Keywords:** stroke, types of strokes, rehabilitation, grip strength, motor functions, stabilization

## Abstract

Almost half of the patients surveyed report impaired function of the upper limbx and handx after stroke. The effect of the passive trunk and shoulder stabilization on the recovery of coordinated hand movement is unclear. This study examined whether passive stabilization of the trunk and shoulder could improve the functional state of the hands after various types of strokes. It is an observational prospective cohort study conducted at the Rehabilitation Clinic in two parallel groups of patients with four different types of strokes (hemorrhagic and ischemic of the brain, similar to the cerebellum). A total of 120 patients were analyzed. Patients were examined in various positions: sitting without a backrest with the upper limb adjacent to the body, supine with the upper limb perpendicular to the body, and supine with the arm stabilized in relation to the patient’s body. Hand Tutor devices and a hand dynamometer were used for the measurements. The frequency and maximum range of motion as well as the grip strength were measured in three different positions of the trunk and upper limb. Passive stabilization of the trunk and shoulder showed more statistically significant differences in Group II. In group II, both in patients after hemorrhagic stroke (wrist Hz *p* = 0.019; wrist ROM *p* = 0.005; Hz F5 *p* = 0.021; Hz F4 *p* = 0.016; Hz F3 *p* = 0.019; Hz F2 *p* = 0.021) and ischemic stroke (*p* = 0.001 for wrist Hz, wrist ROM, Hz F from 5 to F2; and ROM F1; ROM F3 *p* = 0.009; ROM F2 *p* = 0.010), and hemorrhagic cerebellum, improvement of parameters was observed. Stabilization of the upper limb and passive stabilization of the trunk improved the frequency and range of movements in the radiocarpal joint and in the fingers of patients after stroke, regardless of the type of stroke.

## 1. Introduction

Almost half of the stroke patients report impaired function of the affected upper limb and hand [1,2]. This may be related to an insufficient or inadequate physical therapy process. Hayward et al. based on a literature review, reported that upper limb exercises lasted from 4 to 5.7 min, from 23 to 32 repetitions per session, in a hospital setting. Slightly longer, from 11 to a maximum of 17 min per session, during occupational therapy [3]. In their work, researchers often devote time to two-handed coordination, recognizing that two-handed coordination is often impaired after a stroke [4]. Moreover, two-handed coordination is important for the results of the functional assessment of stroke patients [5]. Therefore, when determining the results of the coordination of the movement of the upper limb and hand after stroke, most reports assess the results on the basis of two-handed coordination [6,7,8,9,10,11]. While there is evidence that the pattern of movement is automatic and part of a latent hand–eye program, there is no evidence that upper limb spatial attention is represented on the motor map of the cerebral cortex [12]. Moreover, it has been proved that greater ipsilesional alpha sensorimotor communication is associated with greater motor improvement of the upper limbs after therapy in chronic patients after a stroke [13]. More research is needed to understand the patterns of motor coordination to improve motor skills after strokes [14,15]. It is commonly assumed that for the proper movement of all parts of the body, including the arm and hand, stabilization of the trunk is required and that a stable torso balances the movements of the upper and lower limbs [16,17]. In patients after a stroke, the tension and strength of the stabilizing muscles of the superficial and deep muscles are most often weakened, which leads to asymmetry and abnormal movement patterns [18]. Such disturbances in the motor coordination of the trunk, observed in stroke patients, may make it difficult to restore the motor functions of the hands during physical therapy. Although core stability training has been shown to help in the areas of low back pain and sports [19,20], the role in patients with strokes and neurological deficits is less studied. It is known that precise hand movements depend on proper mobility and positioning of the scapula. In patients after a stroke, shoulder and scapular stabilizers are often weakened [21,22]. In particular, the supraspinatus plays an important role in abduction, flexion, and external rotation in the brachial joint. Scapula stabilization exercises have been shown to improve the function of the affected upper limb in stroke patients [23,24]. The position of the forearm is of particular importance for obtaining a greater grip force. Significantly higher parameters were noted in the transverse position and in the lateral plane, compared to the plane consistent with the body axis and horizontal position [25]. Stabilization is the key to restoring normal movement pattern and improving hand function in patients with impaired motor coordination [24,26,27]. Passive stabilization of the trunk can also help to improve the motor function of the hand. Patients after acute stroke and healthy volunteers whose upper limbs were examined in different starting positions showed that a stable lying position was more favorable than unstable [24,28]. In turn, Yang et al. found that upper limb training with trunk support and ubiquitous feedback helps to improve trunk stability, balance, and upper limb function [29]. El-Nashar et al. concluded that there is no significant difference between training the trunk muscles and a conventional program of physical therapy for the improvement of the upper limbs [30,31]. Moreover, researchers have proved that the movements in the elbow joint depend on the abduction possibility in the shoulder joint. Moreover, the force of shoulder abduction influences the moment of elbow flexion and thus the range of motion in post-stroke patients [24,32,33]. However, it is still difficult to clearly assess what is important for improving the motor coordination of the distal part of the upper limb and whether the type of stroke is important for the recovery of normal motor function.

The aim of the study was to analyze the influence of different positions of the trunk and the affected upper limb on the improvement of hand motor function and handgrip strength in patients after various types of strokes.

## 2. Material and Methods

### 2.1. Study Design

This is an observational, prospective cohort study, the aim of which was to analyze the ranges and frequencies of movements as well as the handgrip strength in patients after various types of strokes, in selected parts of the trunk and affected upper limb. The research was carried out in parallel, two groups of patients with four different types of strokes (hemorrhagic and ischemic of the brain, and similarly of the cerebellum). In the first group of patients, the assessment was performed in a sitting and lying position with the upper limb stabilized against the body. In the second group, the examination was carried out in the supine position with a different position of the upper limb (perpendicular to the body, and then with the stabilized upper limb in relation to the patient’s body). The maximum range of movement (max ROM) and movement frequency (Hz) of wrist and fingers (F), as well as handgrip strength (dependent variables), were collected and then analyzed by comparing the test results in the tested baseline positions (independent variables), for any type of stroke. Then, an analysis was performed in each group, between the different types of strokes.

Inclusion criteria for the stroke group was: (1) patients after stroke (unilateral) and cerebellum stroke; (2) persons with a stable trunk (trunk control test 70–100 points); (3) persons who were functional enough to allow upper limb movements (FMA-UE 40-66 points of motor functions); (4) muscle tone (MAS 0–1+); (5) no serious deficits in communication, memory, or understanding; (6) at least 20 years.

Exclusion criteria for the stroke group: (1) stroke up to five weeks after the episode; (2) another neurological disease; (3) lack of trunk stability; (4) lack of wrist and hand movement; (5) muscle tension (>2 MAS); (6) high or very low blood pressure; (7) severe communication, memory or understanding disorders; dizziness or a malaise of the respondents.

### 2.2. Ethical Approval

The study was carried out in the Department of Rehabilitation of the Military Medical Institute (MMI) in Warsaw, Poland. It was approved by and carried out in accordance with the recommendations of the Ethical Committee of the Military Medical Institute (MMI; approval number 4/MMI/2020). Prior to inclusion, all subjects were informed about the purpose of the study. Written informed consent was obtained from all subjects in accordance with the tenets of the Declaration of Helsinki.

### 2.3. Subjects

160 people were examined. Forty people were excluded (30 stroke patients because of their functional condition, 10 of them declined to participate). The recruitment of patients according to the defined by inclusion/exclusion criteria always consisted of the assessment of the patient by the designated physiotherapist with tests/scales: TCT, FMA-UE, MAS, after examination by the medical doctor/neurologist admitting patients to the clinic. The group post-stroke patients were recruited from the Department of Rehabilitation MMI.

Finally, 120 stroke patients (61 women and 59 men) were included. The group of stroke patients were 5–7 weeks past stroke in the disease, with stable trunk (the Trunk Control Test 74–100 points); subjects were in a functional state allowing movements of the upper extremity (FMA-UE 43–49 motor function points, and normal sensation/light touch); tension of forearm and hand muscles measured with Modified Ashworth Scale (MAS 1/1 +) [34,35,36,37]. Among them, there were 20 patients after a cerebrum hemorrhagic stroke, 80 after a cerebrum ischemic stroke, 10 after a cerebellum hemorrhagic stroke, and the same number after an ischemic cerebellum stroke. Patients of each type of stroke were randomized into Groups I and II. The characteristics of the subjects are shown in Table 1, Table 2 and Table 3.

The flow of participants through each stage of the study is shown below (Figure 1).

### 2.4. Procedure and Measurements

The research was carried out according to the protocol no 5/KRN/2020, registered and published in Clinical Trial Registration.

In two parallel groups of patients after a stroke, the ranges of movements and frequencies and the strength of the handgrip were assessed in different starting positions of the trunk and the affected upper limb. In each group, the first position differed and the second position was the same.

Before each test, the patients were precisely instructed.

In Group I of patients after stroke, the tests were carried out in two starting positions, sitting and lying.

Sitting position. The patients sat on the therapeutic table (without back support), feet resting on the floor. The upper limb was examined in adduction, with the elbow bent in the intermediate position between pronation and supination of the forearm, wrist, and hand without stabilization.Lying position. In the supine position, the upper limb was held beside the subject’s body (adduction in the humeral joint, elbow flexion in the intermediate position, wrist and hand without stabilization).

In Group II of patients after a stroke, the tests were carried out in the supine position with a different position of the upper limb.

Lying position. Supine, with the upper extremity positioned perpendicularly to the trunk.Lying position. The same as in the first group of patients.

A Hand Tutor device was used to measure, composed of a safe and comfortable glove equipped with position and motion sensors (sensitive electro-optical sensors evaluating a position, speed wrist and finger movement; power supply: voltage: 5 [V] DC, rated current input: 300 [mA]), and the Medi TutorTM software, (MediTouch, Tnuvot, Israel,). A Hand Tutor was used to measure the kinematic parameters like the maximum range of movement (ROM) from flexion to extension (sensitivity: 0.05 [mm] of wrist and fingers Ext./Flex) as well as the frequency of movement (motion capture speed: up to 1 [m/s]) [24,38]. Medi Tutor presents data in Excel. The maximum range of motion data during fast frequency estimation movements is represented as ROM.

The system (MediTouch, Tnuvot, Israel) is used by many physical and occupational therapy centers, and has CE and FDA certification [24,28,39]. A manual electronic dynamometer (EH 101) was used for grip strength measurement (Camry, China) (error of measurement, 0.5 kg/lb).

After putting on the glove and setting up the device, the subject was asked to make moves as quickly and in as full a range as possible. The Hand Tutor device recorded at the same time the maximum range of motion during the movement, performed as quickly as possible (max ROM) and the speed/frequency of movements (number of cycles from flexion to extension per unit of time), successively for the wrist and then for the fingers. The duration of the test is imposed and determined by the developers of the device. The assessment of the frequency of movements and the ROM, measured automatically, were performed over time 10 s. The measurement of grip strength with a dynamometer was performed in each position after the maximum range of motion and speed/frequency tests. The upper extremity tested in stroke patients was the paretic extremity.

### 2.5. Sample Size Calculation

Estimating the minimum number of samples was carried out using the G*Power 3.1.9.4 program. The sample size was estimated for a mixed scheme (2 measurements, 4 groups). Assumed α = 0.05; Power = 0.8; F = 0.4 (strong effect). The minimum number of the entire sample is 56 people.

### 2.6. Statistical Analysis

Statistical analyzes were performed using IBM SPSS Statistics 26.0. In order to compare the differences in parameters between the positions used in the study, the Wilcoxon test was performed within one group of patients with a given type of stroke. The Kruskal–Wallis H test was performed to compare patients with hemorrhagic stroke, ischemic stroke, cerebellar hemorrhagic stroke, and cerebellar ischemic stroke in each of the intervention groups. In order to establish the nature of the differences, a post hoc analysis was performed using Dunn’s test with correction of the Bonferroni significance level. The level of significance was α = 0.05.

## 3. Results

For a better understanding of the changes taking place as a result of the interventions, the manuscript includes a supplement. In the supplementary material, we present the initial parameters of passive and active movement ranges of the wrist and fingers (Appendix A). Next, using the Wilcoxon test, the results of coordination and strength were compared by the type of stroke in Group I; that is, the group in which the tests were carried out in sitting and lying positions with the upper limb against the patient’s body. The analysis showed no significant differences in parameters between the positions for patients after hemorrhagic stroke (Table 4), cerebellar hemorrhagic stroke (Table 6), and cerebellar ischemic stroke (Table 7). The only significant differences in parameters occurred among patients after ischemic stroke (Table 5). In the sitting position, they obtained lower results for wrist MaxROM and higher results for fingers 3 and 4 MaxROM. No differences were noted for the remaining parameters.

Then, a comparison was made using the Wilcoxon test in Group II (assessment of coordination and grip strength in the positions lying with the upper limb up and lying with the upper limb against the patient’s body) for each type of stroke. The analysis showed that among hemorrhagic stroke patients, higher results were obtained for Hz of wrist movements, MaxROM of the wrist, HzF5, HzF4, and HzF2 in the supine position with the upper limb against the body than in the supine position with the upper limb up (Table 8). In ischemic stroke patients, higher scores were obtained in the lying position with upper limbs against the body for Hz of the wrist movements, MaxROM of the wrist, HzF5, HzF4, HzF3, MaxROM for F3, HzF2, MaxROM for F2, and MaxROM for F1 (Table 9). In patients after a cerebellar hemorrhagic stroke, higher scores were reported in the lying with upper limb to the body, for HzF1 and MaxROM F1, than in the upper limb in up position (Table 10). In patients after cerebellar ischemic stroke, there were no differences in parameters between items (Table 11).

Finally, the study results were compared between the different types of strokes in each of the analyzed groups.

Group I—in order to compare patients with hemorrhagic stroke, ischemic stroke, cerebellar hemorrhagic stroke, and ischemic stroke in terms of the range of motion, frequency of movements, and hand grip strength, sitting, and lying were analyzed using the H Kruskal–Wallis test. The analysis only showed significant differences between the groups for the MaxROM of the wrist in a sitting position. In order to establish the nature of the differences, a post hoc analysis was performed using the Dunn test with a correction of the Bonferroni significance level. Adjusted for the significance level for multiple comparisons, the study showed no significant differences between the groups with different types of strokes. Detailed results are summarized in Appendix A.

Group II—in order to compare patients with different types of strokes in terms of ranges of motion, frequency of movement, and grip strength, the positions of lying with the upper limb up and lying with the upper limb against the patient’s body were analyzed using the H Kruskal–Wallis test. The analysis only showed significant differences between the groups for the wrist Hz parameter. In the position with the upper limb up and for Hz P1 (finger 1), in the position with the upper limb against the body. In order to establish the nature of the differences, a post hoc analysis was performed using the Dunn test with a correction of the Bonferroni significance level. Among hemorrhagic stroke patients, Hz wrist was lower than in ischemic stroke patients (*p* = 0.033). For Hz P1, after taking into account the correction of the significance level for multiple comparisons, the analysis did not show any significant differences between the groups. Detailed results are summarized in Appendix A.

## 4. Discussion

The results of an observational cohort study showed that the greatest improvement in the wrist and hand range of movement and frequency in stroke patients can be achieved in the supine position with stabilized upper limb. The ranges of movement and frequency were assessed using the HandTutorTM. This device has already been used to test the ranges of passive and active mobility as well as frequency, and maximum ranges of motion of the wrist and hand movements [24,28]. It also appeared in the work of Carmela et al., although in their work the device, apart from the test function, was assessed in terms of the effects of therapy [39]. For the functional assessment of the subjects, commonly accepted scales and tests, the Trunk Control Test and the Fugl–Meyer score, as well as the Modified Ashworth Scale were used [34,35,37].

The comparison of the results of frequency, ranges of movement, and strength depending on the type of stroke in the studied groups of patients showed less significant results in Group I; that is, in the group in which the tests were carried out in the sitting and lying positions with the upper limb against the patient’s body. In this case, stabilization of the upper limb against the patient’s body in the supine position resulted in a significant improvement in the result only for the wrist maxROM. On the other hand, in the unstable position, sitting in the same group of patients after ischemic stroke, significantly higher results were obtained for the max ROM of fingers 3 and 4. Much more significant results were recorded in Group II. The research was carried out there, in the lying position with different positions of the upper limb, perpendicular to the body and close to the patient’s body. In this group, both after hemorrhagic and ischemic stroke and after a cerebellar hemorrhagic stroke, more statistically significant results were noted in the lying position with stabilization of the upper limb against the patient’s body. After the cerebral hemorrhagic stroke, higher frequency scores for both wrist and fingers from 2 to 5 and wrist Max ROM were obtained. Similarly, after ischemic stroke, higher results were obtained in the lying position with the upper limb against the body for the wrist and fingers frequency, from 2 to 5, and the max ROM of the wrist and fingers 3, 2, and 1. In turn, after cerebellar hemorrhagic stroke in a stable position, higher results for Hz F1 and MaxROM F1 were recorded. Only after cerebellar ischemic stroke were no differences in parameters between assessing positions noted.

Our tests, carried out to assess the position of the trunk and upper limb for the parameters of a range of movement, frequency, and handgrip strength, indicate the superiority of the position lying with a stabilized upper limb against the body, regardless of the type of stroke we are dealing with.

Comparing patients with different types of strokes in group I, the analysis with the Kruskal–Wallis H test showed significant differences for the MaxROM of the wrist in a sitting position. Still, the post hoc analysis with the Dunn test, with a correction of the Bonferroni significance level, showed no significant differences between patients with different types of strokes. The analogous analysis in Group II showed significant differences only for the Hz wrist with the upper limb up and for Hz F1 with the upper limb against the body. Post hoc analysis with the Dunn test with correction of the Bonferroni significance level showed that in patients after a hemorrhagic stroke, Hz wrist values were lower than in patients after ischemic stroke. However, for Hz F1, the analysis showed no significant differences. The obtained result may be a consequence of unequal numbers of cases with different types of strokes as well as the type of stroke and the functional state resulting from this. However, taking into account the characteristics of the patients on the exit, the greater likelihood of obtaining similar results is on the side of the size of the groups with different types of strokes. Moreover, by looking at the biometric characteristics and baseline motor parameters of patients in groups I and II (similar groups of patients), we can see more significant differences in group II as a result of the intervention. We speculate that it is the upper limb position, not the torso position, that is important for achieving better frequency and range of motion scores on the distal portion of the affected upper limb. Moreover, analysis of the results of the handgrip strength tests of the affected upper limb in each of the selected body positions showed no significant improvement in any of the studied patients.

It has previously been shown that the strength and position of the forearm affect the activity and strength of the upper limb [40,41]. For example, de Ponte et al. found that greater grip hand strength was generated with the forearm turned inward. In this position, the flexors and extensors of the hand and wrist showed greater potential in this position [41].

We speculate that in our work, this may be due to both the items taken for measurement and, primarily to the baseline values of muscle tone (reduced from 0 to 1+ on the Ashworth scale) and muscle strength after stroke in the acute phase.

Previous studies have shown that upper limb task training with abdominal cramps (i.e., active stabilization of the trunk) improved gait and balance in patients with hemiparesis [42,43]. On the other hand, the influence of passive trunk and shoulder stabilization on hand and wrist movement has been rarely analyzed. In the introduction to the work, the authors present those who have previously dealt with this subject.

Thus, the first of them, Souque, in 1916 describes the phenomenon of the fingers of a hand straightening when passively lifting the arm up. Subsequently, Brunnstrom in his book emphasizes the importance of this phenomenon [32]. In 2010, Nijland et al. reported a study showing that recovery of hemiplegic arm function after 6 months can be predicted in a hospital stroke unit using two simple tests: finger extension and arm abduction [44].

The aim of our research was to assess the position of the body and upper limb to obtain significant results in frequency, range of motion, and handgrip strength, as well as to check whether the type of stroke is important in obtaining the results.

Other researchers have found that shoulder stabilization exercises improve handgrip strength in patients with shoulder tension syndrome, and shoulder stabilization exercises improve the function of upper limb paresis in stroke patients [23,45]. Similarly, in a study of children with cerebral palsy, it was found that neck and trunk stabilization exercises improve hand function in writing, turning pages, inserting small objects into something, lifting large and light cans, and lifting large and heavy cans [46]. Various factors can also affect the coordinated movement of the hand and wrist. For example, Lee et al., also Morrison et al., found that an increase in the frequency of movements disturbs the coordination of the upper limb, and that the speed of movement during clapping disturbs coordination [47,48]. Similarly, noise or issuing commands may disrupt two-handed coordination [49,50]. These factors should be considered in any rehabilitation program aimed at improving the function of the upper limb and hand after a stroke. Moreover, passive stabilization of the trunk (or immobilization of the trunk) may allow stroke patients to access “normal” movement patterns lost due to neurological diseases [27]. Our work shows that even passive stabilization of the trunk and upper limb, also with reduced tension and lower muscle strength, as in patients after a stroke, can affect the parameters of hand movement and allow stroke patients access to hidden movement patterns during rehabilitation.

### 4.1. Research Value

Placing the patient in a supine position, and holding the paresis-affected arm close to the body to stabilize the shoulder, may be useful during physiotherapy tasks to improve the function and activity of the distal paralysis limb after a stroke.

### 4.2. Study Limitation

The main limitation of the study was the number of people tested in both groups for each type of stroke. However, the authors tried to match patients in terms of biometrics and epidemiology, and in terms of the number of people with each type of stroke between groups. Moreover, we decided to make such a comparison, assuming that k + 1 observations (where k is the number of compared groups) are enough to compare. In our case, there is a minimum of 5 people in the group. Certainly, in future studies, the authors will want to analyze equal and more numerous groups of patients with different types of strokes. Another limitation of the study is the fact that the patients were examined in a functional state that allowed movement (e.g., muscle tension MAS 1/1+), on the other hand, in order to assess the frequency, range of movements, and grip strength, it is a functional state that is essential for any movement of the wrist and fingers. We believe it would be good to study functionally different groups of patients.

## 5. Conclusions

Stabilization of the upper limb and passive stabilization of the trunk improves the coordination of movements in the radiocarpal joint and hands of patients after a stroke, regardless of the type of stroke.Passive stabilization of the trunk and upper limb against the subject’s body is very important for regaining precise, coordinated movements of the distal part of the upper limb.

## Figures and Tables

**Figure 1 brainsci-12-01234-f001:**
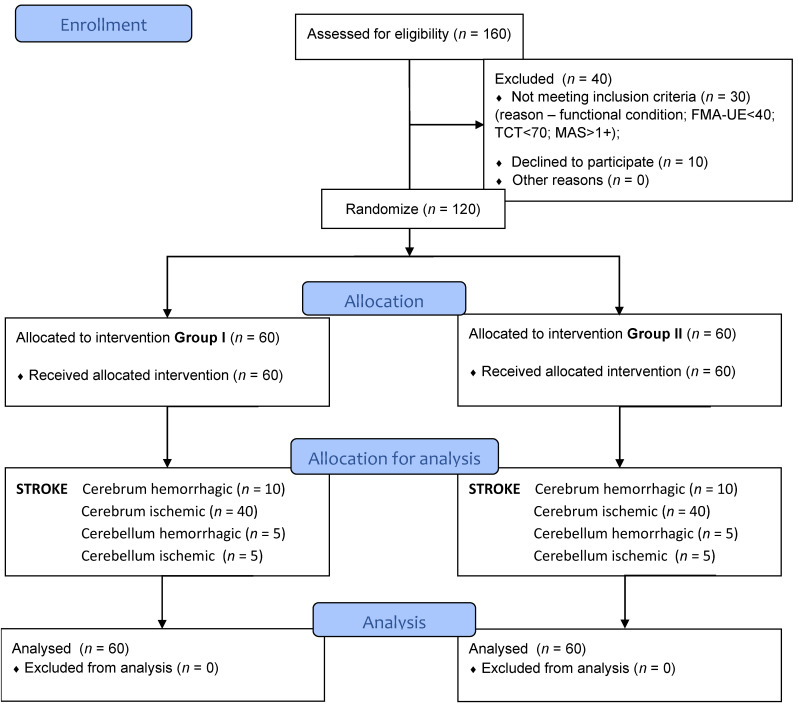
Flow of participants through each stage of the study.

**Table 1 brainsci-12-01234-t001:** Biometric data of the Group I—first post-stroke study population.

Group I		Age		Body Mass		Height		BMI	
	N	M	SD	M	SD	M	SD	M	SD
Cerebrum hemorrhagic stroke	10	65.00	10.45	76.80	8.01	170.20	4.61	26.46	1.96
Cerebrum ischemic stroke	40	66.73	14.47	74.63	9.12	171.15	9.75	25.44	1.83
Cerebellum hemorrhagic stroke	5	53.80	23.23	77.60	15.85	168.00	5.83	27.43	5.17
Cerebellum ischemic stroke	5	66.40	10.55	84.00	9.14	173.20	4.82	27.95	2.11
H		1.97		4.42		1.74		7.24	
P		0.578		0.220		0.629		0.065	
η^2^		<0.01		0.01		<0.01		0.04	

**Table 2 brainsci-12-01234-t002:** Biometric data of the Group II—second post-stroke study population.

Group II		Age		Body Mass		Height		BMI	
	N	M	SD	M	SD	M	SD	M	SD
Cerebrum hemorrhagic stroke	10	66.00	20.93	72.90	17.90	164.30	9.12	26.75	4.59
Cerebrum ischemic stroke	40	63.33	14.87	77.97	15.98	170.93	10.31	26.51	4.18
Cerebellum hemorrhagic stroke	5	46.60	22.99	86.80	20.97	174.20	10.69	28.79	7.90
Cerebellum ischemic stroke	5	53.60	21.08	80.80	17.22	167.60	8.76	29.08	7.58
H		4.59		4.88		4.91		2.33	
P		0.205		0.181		0.179		0.508	
η^2^		0.01		0.01		0.02		<0.01	

**Table 3 brainsci-12-01234-t003:** Epidemiological data of post-stroke populations.

Total Number of Patients *n* = 120 (100%)	Post-Stroke Group I *n* = 60 (50%)	Post-Stroke Group II *n* = 60 (50%)
Female	30 (50%)	31 (51.67%)
Male	30 (50%)	29 (48.33%)
Cerebrum hemorrhagic stroke (unilateral subcortical)	10 (16.67%)	10 (16.67%)
Cerebrum ischemic stroke (unilateral subcortical)	40 (66.67%)	40 (66.67%)
Cerebellum hemorrhagic stroke	5 (8.33%)	5 (8.33%)
Cerebellum ischemic stroke	5 (8.33%)	5 (8.33%)
Time post-stroke/episode (weeks)	5–7	5–7
Right affected side	30 (50%)	36 (60%)
Left affected side	30 (50%)	24 (40%)
Dominant right hand	60 (100%)	60 (100%)
TCT (points 48–61) ± SD	81.6 ± 9	80.38 ± 11
FMA-UE (points 43–49) ± SD	45.45 ± 7.72	45.73 ± 9.03
MAS (degrees 0/1/1+) (examined *n*)	0/1/1+ 0/32/28	0/1/1+ 0/40/20

**Table 4 brainsci-12-01234-t004:** Comparison of movement and strength parameters in sitting and lying positions with upper limb against the body in patients after a hemorrhagic stroke.

	Sitting Position			Lying, Upper Limb Stabilized					
Parameters	M	Me	SD	M	Me	SD	Z	p	r
Hz wrist [cyc/s]	1.25	1.05	0.87	1.56	1.50	1.15	−1.58	0.113	0.35
Wrist MaxROM [mm]	21.46	19.45	13.60	22.58	20.25	14.20	−1.38	0.169	0.31
Hz F5	1.87	1.95	0.85	2.05	1.85	0.71	−1.26	0.206	0.28
MaxROM F5	17.25	18.20	9.18	19.78	19.05	14.61	−0.36	0.721	0.08
Hz F4	1.86	1.95	0.84	2.08	1.85	0.4	−1.78	0.075	0.40
MaxROM F4	22.13	22.90	9.08	21.47	22.40	7.35	−1.07	0.284	0.24
Hz F3	1.85	2.00	0.88	2.08	1.85	0.74	−1.70	0.090	0.38
MaxROM F3	22.43	23.95	7.51	20.58	21.30	5.34	−1.38	0.169	0.31
Hz F2	1.85	2.00	0.88	2.08	1.85	0.74	−1.70	0.090	0.38
MaxROM F2	18.22	20.00	6.79	16.85	17.85	5.44	−0.77	0.444	0.17
Hz F1	1.72	2.00	1.01	1.64	1.80	1.15	0.00	1.000	0.00
MaxROM F1	11.67	12.10	5.23	9.01	10.15	4.47	−1.38	0.169	0.31
Grip strength [kg]	19.82	14.95	17.02	20.38	13.95	17.14	−1.48	0.139	0.47

Legend: M—mean; ROM—range of motion from flexion to extension; SD—standard deviation; Wilcoxon test; one cycle = the movement from flexion to extension.

**Table 5 brainsci-12-01234-t005:** Comparison of movement and strength parameters in sitting and lying positions with upper limb against the body in patients after ischemic stroke.

	Sitting Position			Lying, Upper Limb Stabilized					
Parameters	M	Me	SD	M	Me	SD	Z	p	r
Hz wrist [cyc/s]	1.05	0.80	0.65	1.00	0.85	0.65	−0.51	0.609	0.06
Wrist MaxROM [mm]	14.03	14.00	5.37	19.06	21.10	6.37	−4.43	0.001	0.50
Hz F5	1.45	1.15	0.93	1.54	1.25	1.00	−1.43	0.154	0.16
MaxROM F5	17.36	16.65	8.01	16.20	15.60	7.79	−1.52	0.129	0.17
Hz F4	1.43	1.15	0.95	1.49	1.15	1.01	−0.94	0.347	0.11
MaxROM F4	21.26	21.10	7.81	18.87	18.85	6.24	−3.20	0.001	0.36
Hz F3	1.44	1.15	0.93	1.54	1.25	1.00	−1.45	0.147	0.16
MaxROM F3	20.29	19.75	4.95	19.19	19.50	5.04	−2.08	0.037	0.23
Hz F2	1.45	1.15	0.92	1.54	1.25	1.00	−1.40	0.163	0.16
MaxROM F2	16.85	17.60	4.81	16.50	15.90	4.80	−1.09	0.276	0.12
Hz F1	1.11	0.90	0.89	1.04	0.90	0.78	−0.45	0.654	0.05
MaxROM F1	8.13	7.25	5.34	7.48	7.15	4.86	−1.49	0.137	0.17
Grip strength [kg]	17.25	15.05	11.38	18.13	15.55	11.53	−1.30	0.192	0.15

Legend: M—mean; ROM—range of motion from flexion to extension; SD—standard deviation; Wilcoxon test; one cycle = the movement from flexion to extension.

**Table 6 brainsci-12-01234-t006:** Comparison of movement and strength parameters in sitting and lying positions with upper limb against the body in patients after cerebellar hemorrhagic stroke.

	Sitting Position			Lying, Upper Limb Stabilized					
Parameters	M	Me	SD	M	Me	SD	Z	p	r
Hz wrist [cyc/s]	1.64	1.50	0.84	1.32	1.20	0.90	−0.81	0.416	0.26
Wrist MaxROM [mm]	19.10	19.20	1.01	20.60	21.00	2.65	−0.81	0.416	0.26
Hz F5	1.54	1.30	0.72	1.84	1.50	0.79	−1.22	0.223	0.39
MaxROM F5	16.80	21.20	9.79	15.24	18.20	7.52	−0.41	0.686	0.13
Hz F4	1.54	1.30	0.72	1.86	1.50	0.83	−1.22	0.223	0.39
MaxROM F4	21.14	21.80	6.14	21.04	18.60	9.57	−0.14	0.893	0.04
Hz F3	1.42	1.00	0.81	1.84	1.50	0.79	−1.76	0.078	0.56
MaxROM F3	21.52	22.50	4.97	21.24	20.20	6.33	−0.14	0.893	0.04
Hz F2	1.42	1.00	0.81	1.86	1.50	0.83	−1.76	0.078	0.56
MaxROM F2	20.60	20.10	6.00	20.28	19.60	5.62	−0.41	0.684	0.13
Hz F1	1.38	1.00	0.94	1.42	1.20	1.27	−0.14	0.892	0.04
MaxROM F1	11.20	10.00	8.87	11.94	13.50	6.59	−0.67	0.500	0.21
Grip strength [kg]	25.32	22.70	18.96	26.20	22.70	17.47	−0.73	0.465	0.23

Legend: M—mean; ROM—range of motion from flexion to extension; SD—standard deviation; Wilcoxon test; one cycle = the movement from flexion to extension.

**Table 7 brainsci-12-01234-t007:** Comparison of movement and strength parameters in sitting and lying positions with upper limb against the body in patients after cerebellar ischemic stroke.

	Sitting Position			Lying, Upper Limb Stabilized					
Parameters	M	Me	SD	M	Me	SD	Z	p	r
Hz wrist [cyc/s]	1.20	0.70	0.87	1.02	0.70	0.72	−0.92	0.357	0.29
Wrist MaxROM [mm]	20.72	21.70	4.18	23.60	22.00	4.61	−1.21	0.225	0.38
Hz F5	1.28	0.80	0.90	1.46	0.90	1.10	−1.47	0.141	0.47
MaxROM F5	22.90	25.30	6.84	22.02	23.00	7.87	−0.67	0.500	0.21
Hz F4	1.28	0.80	0.90	1.44	0.80	1.11	−1.30	0.194	0.41
MaxROM F4	27.64	26.60	4.64	23.62	25.50	4.47	−1.75	0.080	0.55
Hz F3	1.28	0.80	0.90	1.44	0.80	1.11	−1.30	0.194	0.41
MaxROM F3	24.86	25.30	4.54	22.08	20.00	3.58	−1.75	0.080	0.55
Hz F2	1.28	0.80	0.90	1.44	0.80	1.11	−1.30	0.194	0.41
MaxROM F2	21.48	21.90	3.20	20.08	19.20	3.51	−1.75	0.080	0.55
Hz F1	1.28	0.80	0.90	1.44	0.80	1.11	−130	0.194	0.41
MaxROM F1	10.66	9.60	5.64	8.34	9.20	3.85	−0.73	0.465	0.23
Grip strength [kg]	23.28	18.40	12.64	21.44	16.70	13.66	−0.94	0.345	0.30

Legend: M—mean; ROM—range of motion from flexion to extension; SD—standard deviation; Wilcoxon test; one cycle = the movement from flexion to extension.

**Table 8 brainsci-12-01234-t008:** Comparison of the parameters of movement and strength in the lying positions with the upper limb and against the body in the group of patients after hemorrhagic stroke.

	Lying, Upper Limb Perpendicular			Lying, Upper Limb Stabilized					
Parameters	M	Me	SD	M	Me	SD	Z	p	r
Hz wrist [cyc/s]	0.77	0.70	0.49	1.31	1.30	0.75	−2.35	0.019	0.53
Wrist MaxROM [mm]	18.42	19.60	3.88	23.10	22.35	4.73	−2.80	0.005	0.63
Hz F5	1.30	1.00	0.98	1.68	1.50	1.10	−2.30	0.021	0.51
MaxROM F5	16.45	18.20	6.82	15.64	16.30	5.38	−0.87	0.386	0.19
Hz F4	1.29	1.00	0.99	1.69	1.55	1.10	−2.40	0.016	0.54
MaxROM F4	19.24	17.75	5.71	18.38	15.85	6.72	−0.87	0.386	0.19
Hz F3	1.29	1.00	0.99	1.68	1.50	1.10	−2.35	0.019	0.53
MaxROM F3	20.29	21.15	3.51	19.48	19.05	3.78	−0.87	0.385	0.19
Hz F2	1.30	1.00	0.98	1.68	1.50	1.10	−2.30	0.021	0.51
MaxROM F2	16.50	16.85	3.60	16.63	16.85	4.05	−0.05	0.959	0.01
Hz F1	1.29	1.00	0.98	0.82	0.70	0.81	−1.13	0.260	0.25
MaxROM F1	6.88	5.95	5.88	8.22	5.95	8.19	−0.65	0.515	0.15
Grip strength [kg]	19.86	17.15	12.65	18.60	19.40	8.62	−0.05	0.959	0.01

Legend: M—mean; ROM—range of motion from flexion to extension; SD—standard deviation; Wilcoxon test; one cycle = the movement from flexion to extension.

**Table 9 brainsci-12-01234-t009:** Comparison of movement and strength parameters in the lying positions with upper limb and against the body in the group of patients after ischemic stroke.

	Lying, Upper Limb Perpendicular			Lying, Upper Limb Stabilized					
Parameters	M	Me	SD	M	Me	SD	Z	p	r
Hz wrist [cyc/s]	1.53	1.45	0.90	1.84	1.60	0.99	−3.99	0.001	0.36
Wrist MaxROM [mm]	17.28	18.00	8.12	20.01	21.05	6.60	−4.48	0.001	0.41
Hz F5	1.75	1.80	1.03	2.12	2.20	1.10	−3.93	0.001	0.36
MaxROM F5	14.67	15.30	6.61	15.67	15.40	8.51	−0.56	0.576	0.05
Hz F4	1.80	1.85	1.03	2.16	2.15	1.04	−4.19	0.001	0.38
MaxROM F4	18.48	18.70	9.31	18.88	18.90	8.66	−0.49	0.622	0.05
Hz F3	1.81	1.85	1.01	2.21	2.20	1.02	−4.42	0.001	0.40
MaxROM F3	17.88	18.25	6.16	19.34	19.25	6.69	−2.59	0.009	0.24
Hz F2	1.79	1.85	1.03	2.07	2.10	1.06	−3.34	0.001	0.31
MaxROM F2	14.79	15.65	6.79	15.70	16.80	5.62	−2.58	0.010	0.24
Hz F1	1.24	1.00	1.04	1.37	0.95	1.17	−1.53	0.127	0.14
MaxROM F1	6.72	6.35	4.00	8.22	8.35	4.34	−4.58	0.001	0.42
Grip strength [kg]	20.00	19.15	12.48	20.59	19.10	11.33	−0.95	0.340	0.09

Legend: M—mean; ROM—range of motion from flexion to extension; SD—standard deviation; Wilcoxon test; one cycle = the movement from flexion to extension.

**Table 10 brainsci-12-01234-t010:** Comparison of the movement and strength parameters in the lying positions with upper limb upwards and against the body in the group of patients after a cerebellar hemorrhagic stroke.

	Lying, Upper Limb Perpendicular			Lying, Upper Limb Stabilized					
Parameters	M	Me	SD	M	Me	SD	Z	p	r
Hz wrist [cyc/s]	1.78	1.90	0.91	1.94	2.20	0.67	−0.55	0.581	0.17
Wrist MaxROM [mm]	18.50	18.50	2.60	17.80	17.00	3.67	−0.92	0.357	0.29
Hz F5	1.72	1.50	0.71	1.98	1.80	0.58	−1.84	0.066	0.58
MaxROM F5	19.66	20.10	4.17	20.12	19.50	3.07	0.00	1.000	0.00
Hz F4	1.72	1.50	0.71	1.98	1.80	0.58	−1.84	0.066	0.58
MaxROM F4	22.20	21.80	4.99	22.42	19.90	5.65	0.00	1.000	0.00
Hz F3	1.72	1.50	0.71	2.00	1.80	0.57	−1.83	0.068	0.58
MaxROM F3	22.08	21.90	3.72	24.44	26.20	6.51	−1.21	0.225	0.38
Hz F2	1.72	1.50	0.71	2.10	1.90	0.52	−1.83	0.068	0.58
MaxROM F2	17.66	17.90	1.64	20.64	21.70	4.79	−1.48	0.138	0.47
Hz F1	1.48	1.50	1.01	2.00	1.80	0.57	−2.03	0.042	0.64
MaxROM F1	8.24	7.40	2.25	9.46	9.50	2.43	−2.02	0.043	0.64
Grip strength [kg]	16.02	19.10	6.45	17.14	19.70	5.59	−0.94	0.345	0.30

Legend: M—mean; ROM—range of motion from flexion to extension; SD—standard deviation; Wilcoxon test; one cycle = the movement from flexion to extension.

**Table 11 brainsci-12-01234-t011:** Comparison of movement and strength parameters in the lying positions with upper limb upwards and against the body in the group of patients after cerebellar ischemic stroke.

	Lying, Upper Limb Perpendicular			Lying, Upper Limb Stabilized					
Parameters	M	Me	SD	M	Me	SD	Z	p	r
Hz wrist [cyc/s]	1.20	1.20	0.56	1.48	1.60	0.69	−1.60	0.109	0.51
Wrist MaxROM [mm]	16.20	15.80	4.90	19.28	16.00	6.71	−1.46	0.144	0.46
Hz F5	1.26	1.30	0.47	1.58	1.90	0.72	−1.46	0.144	0.46
MaxROM F5	15.94	16.10	6.57	15.60	18.70	8.00	−0.14	0.892	0.04
Hz F4	1.24	1.20	0.59	1.20	1.20	0.86	−0.37	0.715	0.12
MaxROM F4	17.94	16.80	5.88	18.82	20.00	8.94	−0.41	0.686	0.13
Hz F3	1.18	1.20	0.50	1.56	1.90	0.70	−1.84	0.066	0.58
MaxROM F3	18.34	18.40	4.82	18.40	20.80	7.18	−0.67	0.500	0.21
Hz F2	1.26	1.30	0.47	1.48	1.60	0.72	−1.22	0.223	0.39
MaxROM F2	15.10	14.80	4.17	16.70	18.90	6.16	−0.94	0.345	0.30
Hz F1	0.74	0.50	0.51	0.32	0.40	0.16	−1.10	0.273	0.35
MaxROM F1	6.78	4.60	3.71	11.38	13.30	7.36	−1.48	0.138	0.47
Grip strength [kg]	15.84	14.50	7.82	17.04	17.30	7.05	−0.94	0.345	0.30

Legend: M—mean; ROM—range of motion from flexion to extension; SD—standard deviation; Wilcoxon test; one cycle = the movement from flexion to extension.

## Data Availability

Data available on request from corresponding author.

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
