# Peer review of "Effect of the Trunk and Upper Limb Passive Stabilization on Hand Movements and Grip Strength Following Various Types of Strokes—An Observational Cohort Study"

_brainsci, 2022, doi:10.3390/brainsci12091234_

Round 1
Reviewer 1 Report (Previous Reviewer 1)
After the adjustments made by the authors, the study shows a better clarity about the method, data processing procedures and results, and with minor editorial revisions, the study is understandable.
For a better understanding of the progressions achieved, the reader would appreciate a graphical presentation that allows to identify in a timeline the progressions of the kinematics of the intervention segments.
Author Response
Manuscript ID: brainsci-1898368
Type of manuscript: Article
Title: Effect of the trunk and upper limb passive stabilization on hand movements and grip strength following various types of strokes - an observational cohort study.
Dear Reviewers,
Thank you very much for the analysis of our manuscript. We really appreciate your comments and indication of fragments that should be corrected and explained. Considering your suggestions, all mistakes were corrected. The introduction of corrections and changes in the text could have caused the numbering of the lines to shift. In order to avoid misunderstandings, changes introduced in the text are marked in green and additionally, the manuscript was sent in the change tracking mode.
Reviewer #1:
Thank you very much for the very quick and thorough analysis of our manuscript.
Regarding the remarks and comments, I am writing back.
The following comments and answers:
Comments and Suggestions for Authors
After the adjustments made by the authors, the study shows a better clarity about the method, data processing procedures and results, and with minor editorial revisions, the study is understandable.
For a better understanding of the progressions achieved, the reader would appreciate a graphical presentation that allows identifying in a timeline the progressions of the kinematics of the intervention segments.
Indeed, a graphic presentation is a good way to present the results of your work. Following your suggestion, such a presentation was added to the manuscript.
Thank you very much for the suggestion.
Thank you very much for your time.
Reviewer 2 Report (Previous Reviewer 2)
Thanks for all those deep modifications - I feel the paper really improved.
Just one point maybe, more a formatting issue, but I have issues in the pdf document from page 16 to page 25 - those pages are white and I don't see any content (former table 12).
Author Response
Manuscript ID: brainsci-1898368
Type of manuscript: Article
Title: Effect of the trunk and upper limb passive stabilization on hand movements and grip strength following various types of strokes - an observational cohort study.
Dear Reviewers,
Thank you very much for the analysis of our manuscript. We really appreciate your comments and indication of fragments that should be corrected and explained. Considering your suggestions, all mistakes were corrected. The introduction of corrections and changes in the text could have caused the numbering of the lines to shift. In order to avoid misunderstandings, changes introduced in the text are marked in green and additionally, the manuscript was sent in the change tracking mode.
Reviewer #2:
Thank you very much for the very quick and thorough analysis of our manuscript.
Regarding the remarks and comments, I am writing back.
The following comments and answers:
Comments and Suggestions for Authors
Thanks for all those deep modifications - I feel the paper really improved.
Just one point maybe, more a formatting issue, but I have issues in the pdf document from page 16 to page 25 - those pages are white and I don't see any content (former table 12).
Thank you very much for your appreciation of our manuscript.
I don't understand a bit why such a problem arose since the revised manuscript has 16 pages of references. Indeed, after the correction, the last table is not there. Instead, there is text and supplementary material added. Corrected added text in the results section: "Finally, the study results were compared between the different types of stroke in each of the analyzed groups.
Group I - In order to compare patients with hemorrhagic stroke, ischemic stroke, cerebellar hemorrhagic stroke, and ischemic stroke in terms of the range of motion, frequency of movements, and hand grip strength, sitting, and lying were analyzed using the H Kruskal Wallis test. The analysis showed significant differences between the groups only for the MaxROM of the wrist in a sitting position. In order to establish the nature of the differences, a post hoc analysis was performed using the Dunn test with a correction of the Bonferroni significance level. Adjusted for the significance level for multiple comparisons, the study showed no significant differences between the groups with different types of stroke. Detailed results are summarized in Table S5.
Group II - In order to compare patients with different types of stroke in terms of ranges of motion, frequency of movement, and grip strength, the positions of lying with the upper limb up and lying with the upper limb against the patient's body were analyzed using the H Kruskal Wallis test. The analysis showed significant differences between the groups only for the wrist Hz parameter. In the position with the upper limb up and for Hz P1 (finger 1), in the position with the upper limb against the body. In order to establish the nature of the differences, a post hoc analysis was performed using the Dunn test with a correction of the Bonferroni significance level. Among hemorrhagic stroke patients, Hz wrist was lower than in ischemic stroke patients (p = 0.033). For Hz P1, after taking into account the correction of the significance level for multiple comparisons, the analysis did not show any significant differences between the groups. Detailed results are summarized in Table S6.„
Moreover, as suggested by another reviewer, a graphic design has been added to the manuscript, which I hope will make the text easier to understand.
Thank you for your careful study of the manuscript.
Thank you very much for your time.
Reviewer 3 Report (New Reviewer)
The authors have done a commendable job in revising the manuscript. I will suggest the authors to check for any typos.
I also think that this manuscript should have pictorial representation of the behavioral task (e.g. a cartoon or stick figures), this will help in better outreach among the scientific community.
Author Response
Manuscript ID: brainsci-1898368
Type of manuscript: Article
Title: Effect of the trunk and upper limb passive stabilization on hand movements and grip strength following various types of strokes - an observational cohort study.
Dear Reviewers,
Thank you very much for the analysis of our manuscript. We really appreciate your comments and indication of fragments that should be corrected and explained. Considering your suggestions, all mistakes were corrected. The introduction of corrections and changes in the text could have caused the numbering of the lines to shift. In order to avoid misunderstandings, changes introduced in the text are marked in green and additionally, the manuscript was sent in the change tracking mode.
Reviewer #3:
Thank you very much for the very quick and thorough analysis of our manuscript.
Regarding the remarks and comments, I am writing back.
The following comments and answers:
Comments and Suggestions for Authors
The authors have done a commendable job in revising the manuscript. I will suggest the authors to check for any typos.
I also think that this manuscript should have pictorial representation of the behavioral task (e.g. a cartoon or stick figures), this will help in better outreach among the scientific community.
Thank you very much for your appreciation of our manuscript.
As suggested, also by another reviewer, a graphic design has been added to the manuscript, which I hope will make the text easier to understand.
Moreover, the work has been revised and improved in terms of English.
Thank you very much for this suggestion.
Thank you very much for your time.
Reviewer 4 Report (New Reviewer)
The manuscript by Olczak et al is an observational study of stroke patients undergoing physical rehabilitation. The study looks at patients with either hemorrhagic or ischemic strokes in the cerebellum compared to what I presume to be the cortex (authors should clarify the location and extent of damage for non-cerebellar patients). Parameters including frequency of movement, range of motion and grip strength were assessed for the upper limb while the patients were sitting up right or lying down with the upper limb stabilized against the body. The study is well designed, , thoroughly analyzed and the conclusions the authors claim closely match their data. I only have a few minor points to be addressed in a revision:
1. The language is often hard to read and should be reviewed by a native English speaker.
2. The authors show no significant difference in grip strength, please call this out more clearly in the Discussion.
3. From looking over the Data Tables, it seems to me that the ischemic stroke groups (non-cerebellar) saw profoundly more significant affects between sitting up and lying down. Can the authors perform a statistical test to compare the stroke types (e.g. Table 4 vs. 5 and Table 8 vs. 9). This could be very interesting if it is true and would beg the question of whether these types of strokes somehow more significantly destabilized trunk musculature.
4. It is unclear what the p-values in the Abstract are from. Is this averaged data across all the different stroke types?? Please address in Results as well if so.
Author Response
Manuscript ID: brainsci-1898368
Type of manuscript: Article
Title: Effect of the trunk and upper limb passive stabilization on hand movements and grip strength following various types of strokes - an observational cohort study.
Dear Reviewers,
Thank you very much for the analysis of our manuscript. We really appreciate your comments and indication of fragments that should be corrected and explained. Considering your suggestions, all mistakes were corrected. The introduction of corrections and changes in the text could have caused the numbering of the lines to shift. In order to avoid misunderstandings, changes introduced in the text are marked in green and additionally, the manuscript was sent in the change tracking mode.
Reviewer #4:
Thank you very much for the very quick and thorough analysis of our manuscript.
Regarding the remarks and comments, I am writing back.
The following comments and answers:
Comments and Suggestions for Authors
The manuscript by Olczak et al is an observational study of stroke patients undergoing physical rehabilitation. The study looks at patients with either hemorrhagic or ischemic strokes in the cerebellum compared to what I presume to be the cortex (authors should clarify the location and extent of damage for non-cerebellar patients). Parameters including frequency of movement, range of motion and grip strength were assessed for the upper limb while the patients were sitting up right or lying down with the upper limb stabilized against the body. The study is well designed, , thoroughly analyzed and the conclusions the authors claim closely match their data. I only have a few minor points to be addressed in a revision:
- The language is often hard to read and should be reviewed by a native English speaker.
As suggested, the work has been revised and improved in terms of English.
Thank you so much for paying attention to this.
- The authors show no significant difference in grip strength, please call this out more clearly in the Discussion.
The study investigated parameters such as the frequency of movements, and the maximum range of movements of both the wrist and fingers of the affected upper limb in patients after various types of stroke in different starting positions. Indeed, the research did not show any statistically significant differences in the handgrip strength. Considering this is actually speculation. We believe that this may be due to both the positions taken for the measurement as well as, and perhaps most notably, the baseline values of muscle tension (lowered, 0 to 1+, on the Ashworth scale) and muscle strength in post-stroke patients.
Supplementing the information resulted in the addition of two references:
"40. Hoozemans MJM, Van Dieen JH. Prediction of handgrip forces using surface EMG of forearm muscles. Journal of electromyography and kinesiology. 2005; 15 (4): 358-66.
- De Ponte ÁM, Guirro ECdO, Pletsch AHM, Dibai-Filho AV, Brandino HE, Guirro RRdJ. The forearm positioning changes electromyographic activity of upper limb muscles and handgrip strength in the task of pushing a load cart. Journal of Bodywork and Movement Therapies. 2015 2015/10/01 /; 19 (4): 597-603. "
The change in the order of the references caused by this was of course corrected in the text and in the list of references.
As suggested, I have completed the missing information in the discussion section.
Thank you very much for this suggestion.
- From looking over the Data Tables, it seems to me that the ischemic stroke groups (non-cerebellar) saw profoundly more significant affects between sitting up and lying down. Can the authors perform a statistical test to compare the stroke types (e.g. Table 4 vs. 5 and Table 8 vs. 9). This could be very interesting if it is true and would beg the question of whether these types of strokes somehow more significantly destabilized trunk musculature.
Thank you for this comment.
We cannot compare the stroke types between Groups I and II because they were different interventions, but we did compare the stroke types in each group. The results are presented in supplement Tables S5 and S6. In addition, in the results section, there is the text: "Finally, the study results were compared between the different types of stroke in each of the analyzed groups.
Group I - In order to compare patients with hemorrhagic stroke, ischemic stroke, cerebellar hemorrhagic stroke, and ischemic stroke in terms of the range of motion, frequency of movements, and hand grip strength, sitting, and lying were analyzed using the H Kruskal Wallis test. The analysis showed significant differences between the groups only for the MaxROM of the wrist in a sitting position. In order to establish the nature of the differences, a post hoc analysis was performed using the Dunn test with a correction of the Bonferroni significance level. Adjusted for the significance level for multiple comparisons, the study showed no significant differences between the groups with different types of stroke. Detailed results are summarized in Table S5.
Group II - In order to compare patients with different types of stroke in terms of ranges of motion, frequency of movement, and grip strength, the positions of lying with the upper limb up and lying with the upper limb against the patient's body were analyzed using the H Kruskal Wallis test. The analysis showed significant differences between the groups only for the wrist Hz parameter. In the position with the upper limb up and for Hz P1 (finger 1), in the position with the upper limb against the body. In order to establish the nature of the differences, a post hoc analysis was performed using the Dunn test with a correction of the Bonferroni significance level. Among hemorrhagic stroke patients, Hz wrist was lower than in ischemic stroke patients (p = 0.033). For Hz P1, after taking into account the correction of the significance level for multiple comparisons, the analysis did not show any significant differences between the groups. Detailed results are summarized in Table S6."
The text is marked in green.
Thank you very much for this suggestion.
- It is unclear what the p-values in the Abstract are from. Is this averaged data across all the different stroke types?? Please address in Results as well if so.
Indeed, it is difficult to see the parameters presented in the manuscript abstract. As suggested, I made corrections (underlined in green). These are significant results concerning the parameters tested in group II. The amended text reads:
"Passive stabilization of the trunk and shoulder showed more statistically significant differences in group II. In group II, both in patients after hemorrhagic stroke (wrist Hz p = 0.019; wrist ROM p = 0.005; Hz F5 p = 0.021; Hz F4 p = 0.016; Hz F3 p = 0.019; Hz F2 p = 0.021) and ischemic stroke (p = 0.001 for wrist Hz, wrist ROM, Hz F from 5 to F2; and ROM F1; ROM F3 p = 0.009; ROM F2 p = 0.010), and hemorrhagic cerebellum, improvement of parameters was observed. Stabilization of the upper limb and passive stabilization of the trunk improve the frequency and range of movements in the radiocarpal joint and in the fingers of patients after stroke, regardless of the type of stroke."
Due to the length of the abstract, I have limited myself to the given results.
If anything is still difficult to understand, please point out the mistakes.
Thank you very much for any comments.
Thank you very much for your thorough analysis of our work.
Thank you very much for your time.
This manuscript is a resubmission of an earlier submission. The following is a list of the peer review reports and author responses from that submission.
Round 1
Reviewer 1 Report
For a better understanding of the results achieved and a better interpretation of the improvements obtained after the interventions or clinical processes carried out, it is important to know the baseline or specifically the ROM, muscle activation and control values at the beginning of the study. Having this baseline data can allow a better understanding of the progression or improvements achieved through the use of the tutor and the interventions performed.
Reviewer 2 Report
Dear authors and editor, after I carefully read through the paper entitled “Factors significantly changing the results of motor coordination and handgrip strength in people after various types of strokes: an observational study”, I believe the topic of the paper could be of interest, however the quality of the experiment is not suitable for publication in this current form, and deep modifications have to be performed in my opinion. This study is a prospective observational study, which is common in the field. Please find below some comments or suggestions that hopefully may help to improve the current manuscript.
I actually enjoyed the introduction, very clear and concise with good information, however I have several concerns on the remaining parts of the manuscript.
First of all, I found the title a bit long, and maybe lacking clear information about the research – maybe to get the paper a bit more attractive for readers, you could use 1-2 key results to define the title? (You see something like “passive stabilization during rehab improves movements frequency after stroke”)
Then, the main issue concerns the statistical design, which is not really clear and doesn’t fit (for what I understand) your research design. More specifically, you sample size calculation shows 76 participants, but you only have 60 (although I think the sample size calculation is wrong), so what? You keep 60 or you need more?. Also by definition, you research design involves repeated measures, so the statistical analysis (including sample size estimation) should have a repeated component.
Secondly, I think you are misleading the reader when you speak about “motor coordination”, I do not see any measure of coordination in your experiment, but rather a measure of range of motion. Is this right? If yes, you cannot call the range of motion a measure of “coordination” (coordination has a specific definition). Otherwise how did you measure motor coordination?
Last, I wouldn’t say that English is bad or that the paper is not well written, actually the paper is well written, however I believe that sometimes some translations are not done in the usual way, in the sense that some concepts or sentence are strangely organized, potentially due to a direct translation words for words from polish (maybe). For instance “the wrist Hz movements” it’s not really clear what you mean – are you referring to the movement of the wrist frequency? (which I doubt), or are you referring to the frequency of wrist movements? (which I understand from the methodology) – so maybe this needs some checking of english.
Below, please also find some more specific comments
Abstract:
Line 26-27: what is wrist Hz? No mention of Hz before I think (I guess it’s the frequency)
Line 27: what are those 2 p values, 0.019 and <0.001? I think you need to be clearer about what you are comparing exactly, and for what variables.
Line 28: What is an “item”? a variable?
METHODOLOGY:
- How did you split group I and group II? I guess randomized (from Figure 1), but good to explain it in the methodology I guess.
- Sample size calculation: I think the sample size calculation should be put earlier, when you deal with the participants (I don’t see the point of doing a sample size calculation after you selected your participants?). Also on this sample size calculation, I don’t understand why you used 4 groups? (I think you have only 2), as this is a mixed repeated measure design (between and within subjects). Last, you found 76 participants? So what about your study as you don’t reach this sample size right? You should add participants. I understand you put this in the limitations, but this is not a limitations really – the sample size estimation usually happens before the data collection, s you know how many participants you need.
- Statistical analysis: you statistical design doesn’t seem right – at least you need to justify why you used non parametric tests.
- RESULSTS: How many participants of each stroke characteristics in each group (I and II)? Why are you comparing those 4 groups within group I, then within group II separately?
Specific comments:
Line 124: how did you select the 160 participants? As some of them declined, it means you systematically approached everyone in the center?
Line 124: please replace “including” by ”by inclusion”
Line 124: replace 40 by Forty
Line 150: is the registration completed already? And not published?
Line 208: again, what are the “items” here? The variables? Why using items now?
Line 287: What do you mean by “less significant results”? For what I understand the use of p value tells you directly whether the groups are different, in a binary way, i.e., significant or non-significant. How can you characterise a "less significant" result then, is it significant or not?
Line 338: remove the capital A in “also” and the F on the next line